# Research on Key Technologies of Microarray Chips for Detecting Drug-Resistant Genes in *Helicobacter pylori*

**DOI:** 10.3390/mi15030416

**Published:** 2024-03-21

**Authors:** Hongzhuang Guo, Xiuyan Jin, Hao Zhang, Ping Gong, Xin Wang, Tingting Sun

**Affiliations:** 1School of Physics, Changchun University of Science and Technology, Changchun 130022, China; guohongzhuang5221@163.com (H.G.); 2021200033@mails.cust.edu.cn (X.J.); 2School of Life Science and Technology, Changchun University of Science and Technology, Changchun 130022, China; zhanghao@cust.edu.cn (H.Z.); gp@cust.edu.cn (P.G.)

**Keywords:** microarray chip, drug-resistance detection, finite element analysis, *Helicobacter pylori*

## Abstract

In addressing the detection of drug resistance in *Helicobacter pylori*, we have successfully developed an efficient and highly accurate detection methodology. Initially, we designed and fabricated a microarray chip, which underwent finite element analysis for its optical and thermal characteristics. Ultimately, COC material was chosen as the processing material for the chip, ensuring superior performance. Subsequently, we established a comprehensive detection system and validated its performance. Following that, comparative experiments were conducted for detecting drug resistance in *H. pylori*. The experimental results indicate that our established methodology aligns with the results obtained using the E-test detection kit, achieving a concordance rate of 100%. In comparison to the E-test detection kit, our methodology reduces the detection time to 1.5 h and provides a more extensive coverage of detection sites.

## 1. Introduction

*H. pylori* belongs to the order Campylobacterales, the family Helicobacteraceae, and the genus Helicobacter [1]. Humans are the sole hosts for *H. pylori*. This bacterium is widely distributed worldwide, with over half of the global population testing positive for infection. The prevalence of infection is closely associated with individual economic status, environmental sanitation conditions, and lifestyle habits [2]. Numerous studies have confirmed that *H. pylori* is a major cause of gastritis [3,4,5] and peptic ulcers [6]. It is also closely linked to human gastric cancer [7,8,9,10] and gastric MALT lymphoma [11,12]. The groundbreaking work by Versalovic et al. first identified point mutations in the V region of the 23S rRNA of *H. pylori*, leading to resistance to drugs that target the ribosome [13]. A. Occhialini’s research further substantiated that alterations in this region are responsible for *H. pylori* resistance to clarithromycin [14]. Extensive research on the resistance mechanisms of *H. pylori* has resulted in a wealth of reported resistance mutation sites. The most common genetic mutation sites in the 23S rRNA V region include A2143G [15], A2142G, and A2142C, with A2143G being the most prevalent, followed by A2142G [16] and then A2142C [15]. The average minimum inhibitory concentration (MIC) for A2142G/C is higher than that for A2143G. Additionally, the combined mutations of A2142G/C and A2143G lead to a higher MIC compared to single mutations. In a study of Russian samples, Momynaliev et al. identified the A2144G mutation [17]. Analysis of *H. pylori* samples in the Qingdao region by Su Yanhua and others revealed mutation sites such as T2190C, C2195T, A2223G, as well as A2115G [18], G2141A [18], A2142T [19], A2143C [20], A2146G [21], A2146C [21], A2192G [22], G2204C [22], T2215C [22], and G2224A [23], which are potentially associated with clarithromycin resistance.

There are various methods for detecting drug resistance in *H. pylori*, with common approaches including culture methods, molecular biology methods, antibiotic susceptibility testing, mass spectrometry, and genetic sequencing. This paper focuses on a microarray chip-based method utilizing polymerase chain reaction (PCR) within the molecular biology category. This approach stands out for its simplicity of operation, high sensitivity, and excellent reproducibility in detecting drug-resistant genes in *H. pylori*.

Microarray chips are emerging biotechnologies initially applied in nucleic acid hybridization [24,25]. Their principle is based on the interactions among biomolecules, such as DNA, proteins, and polysaccharides, facilitating qualitative or quantitative detection of specific molecules in target samples. This technology finds applications in drug-resistance detection and antibiotic selection. The escalating danger of microbial resistance is attributed to the overuse of antibiotics, rendering them increasingly ineffective against many pathogens, particularly in the presence of bacterial biofilms. Addressing this concern, Giuseppe Brunetti, Annarita di Toma, Caterina Ciminelli, and other scholars proposed a biosensor for antibiotic selection and biofilm inhibitory detection utilizing optical and electrical detection methods, characterized by its low cost, high sensitivity, and rapidity [26,27,28,29,30]. Studies by G. Yin, W. Wu, Q. Yu, and others demonstrated that compared to traditional tissue culture methods, microarray chips exhibit higher sensitivity, specificity, and accuracy in drug-resistance and antibiotic detection [30,31,32,33]. With the advancement of microarray chip technology, researchers aim to further enhance various performance indicators by studying chip substrate material selection and pretreatment, as well as gel point material synthesis, expecting to improve specificity, sensitivity, and stability from different perspectives. In recent years, microarray chip technology has continuously evolved based on previous research, focusing primarily on chip substrate material selection and pretreatment, as well as gel point material synthesis.

This study emphasizes the selection of substrate materials for microarray chips. Commonly used microarray substrate materials include flat glass [34,35], silicon surfaces [36], or filter membranes [37]. The choice of appropriate materials depends on the specific requirements of the chip, as surface characteristics can impact the density of immobilized biomolecules, thereby influencing the sensitivity and specificity of microarray technology [38]. Polymer coatings, often referred to as three-dimensional chemistry, are particularly noteworthy for forming uniform coatings with enhanced binding capabilities, distributing probes in axial positions, and thereby imparting greater selectivity to the recognition of biomolecules associated with the target. Another advantage of polymer coatings is that their structure can be customized for specific applications [39].

In summary, the thin film substrate for microarray chips should meet the following requirements: low excitation effects, low reflectance, the ability to withstand PCR temperatures, and ease of processing. Currently, microarray chips have not had a significant impact on clinical practice comparable to real-time fluorescence quantitative PCR. Some reasons for this include poor reproducibility, high costs, complex workflows (including sample preparation), user subjectivity, and numerous associated technical and non-technical issues even in developed countries and centralized testing laboratories [23,40,41,42]. Therefore, for the detection of drug-resistant genes in *H. pylori*, establishing an efficient, highly sensitive, user-friendly, and easily producible molecular biology detection method is of significant importance for guiding clinical drug usage.

## 2. Materials and Methods

### 2.1. Reagents and Instruments

Electronic analytical balance (METTLER TOLEDO, Greifensee, Switzerland), PCR instrument (Bio-Rad, Hercules, CA, USA), ultrasonic homogenizer (Ningbo Xinzhi, Ningbo, China), vortex mixer (Tiangen Biotech (Beijing) Co., Ltd., Beijing, China), UV curing lamp 312nm UVB, benchtop multimeter (RIGOL MD3058, Beijing, China), oscilloscope (MSOX2002A, Keysight, Colorado Springs, CO, USA), imaging camera (MV-EM200M, Microvision, Xi’an, China), cycloolefin copolymer (TOPAS 5031, Raunheim, Germany), photoresist SU-8 (Micro Chem 2050, Round Rock, TX, USA), methacrylic anhydride (Sigma, St. Louis, MA, USA), glycerol (China National Pharmaceutical Group, Shanghai, China), PB buffer solution (pH 7.25, 0.035 mol/L), SSPE buffer solution, N-(2-Hydroxyethyl)acrylamide (Aladdin, Dubai, United Arab Emirates), *H. pylori* culture medium (Liquid, Tokyo, Japan) and (Qingdao High-tech Industrial Park Haibo Biotechnology Co., Ltd., Qingdao, China).

### 2.2. Source of Samples

The samples were sourced from the First Hospital of Jilin University, with a total of 19 cases. These samples consisted of *H. pylori* isolated from the gastric antrum or corpus tissues of patients resistant to clarithromycin. Confirmation of the presence of *H. pylori* in all isolates was achieved through Gram staining under a microscope, urease, catalase, and oxidase tests. Genomic DNA extraction of *H. pylori* was performed using the Rapid Bacterial Genomic DNA Isolation Kit (Bioteke Corporation, B518225, Shanghai, China). DNA concentration was measured using a NanoDrop 2000 UV-Vis spectrophotometer (Thermo Scientific, Waltham, MA, USA) and adjusted to 100 ng/μL as the corresponding *H. pylori* DNA sample.

### 2.3. Chip Design

The microarray chip design was conducted using SolidWorks(R) Premium 2021 software, as illustrated in Figure 1. The overall dimensions of the chip are 40.8 mm × 10 mm × 0.7 mm. The left side incorporates sample injection holes, while the right side features exhaust holes. The chip adopts a three-layer sandwich structure consisting of a substrate, a cavity layer, and a top layer. The thicker the material thickness, the greater the effect on the transmittance. In order to ensure that the material has a certain anti-deformation strength under the premise that the thickness of the bottom and top sheet should be as low as possible, in this paper the thicknesses of the bottom sheet, cavity sheet, top sheet bit were 0.2 mm, 0.3 mm and 0.2 mm respectively. The morphology of the microfluidic system is unaffected by the temperature field and channel shape. The velocity field fully develops into laminar flow, with a transition zone length of approximately 10.0 mm. The transition zone length for the temperature field to reach a steady state is around 6.5 mm. The inlet hole serves as the injection point for the reaction liquid, and the reaction chamber represents the reaction position of the microarray hydrogel unit. The hydrogel is fixed in the central region, allowing flexible design of array layouts based on different project requirements. Additionally, if needed, an absorbent paper can be added at the far-right exhaust hole to terminate the reaction and eliminate background signals.

### 2.4. Detection System

The system employs Osram LB CRBP blue light-emitting diodes (LEDs) as the light source, and a lens group consisting of two lenses is used to collimate the light source. The axis passing through the light source and the center of the lens group is referred to as the excitation axis. Image acquisition is performed using a lens group and the MV-EM200M imaging camera(Microvision, Xi’an, China), with the axis passing through the lens group and the center of the camera referred to as the detection axis. The device arrangement follows a 30° angle between the two axes, as depicted in Figure 2a. For convenient scanning and detection of the chip, a two-dimensional moving platform is constructed using Shenzhen MicroMI Technology Co., Ltd.’s (Shenzhen, China) BSC5-5-100-BL-L42-C4 linear slide module and Suzhou Rongmai Technology Co., Ltd.’s (Suzhou, China) STC-42D20-0461 stepper motor, as shown in Figure 2b.

### 2.5. Primers and Probes

Primers and probes for the identification of *H. pylori* and the detection of resistance mutation sites in the 16S rRNA, cytotoxin-associated gene A (CagA, GenBank: GQ161098.1), and 23S rRNA V region were designed based on the sequences of the standard strain of *H. pylori* (*H. pylori* strain 26695). All primers and probes were synthesized by Sangon Biotech (Shanghai) Co., Ltd., Shanghai, China, as shown in Table 1.

### 2.6. Preparation of Microarray Chips

The microarray chip utilizes hydrogel as the reaction component. Preparation of the hydrogel involves mixing the configured system (20% HEMAm; 1% bis; 65% glycerol; 14% PB buffer solution; 200 pmol/μL corresponding probe) and injecting it into the spotter. The spotting is performed on the substrate material of the chip and then subjected to UV cross-linking and curing under conditions of 50 °C and 312 nm UVB exposure for 30 min. The printed gel array is cleaned in 1× SSPE buffer solution for 1 h, followed by multiple rinses with MilliQ water and air-drying before use.

### 2.7. Experimental Section

#### 2.7.1. Experimental Investigation of Substrate Materials for Microarray Chips

Four materials meeting the simulation results from COMSOL Multiphysics 5.4 were chosen to receive excitation light at 497 nm, and the excitation effects were measured. Using the chip material as a variable, a control experiment was conducted. Four materials were selected as the bottom and top layers of the chip. Using the fluorescence signal acquisition system, chips made of the four materials were exposed for 50 s under 494 nm excitation light. Sixteen-bit images were captured, and the ImageJ 1.52d image analysis software was used to extract the signal and calculate the fluorescence mean value of the chip material. Due to the different reaction effects of the excitation light on the microdroplet-loaded hollow chambers and the chip edges connected by gel from the top and bottom layers, grayscale values of the chambers and edges in the images were separately collected for statistical analysis to determine the appropriate chip material.

#### 2.7.2. Verification of Optical Uniformity in the Fluorescence Signal Acquisition System

To ensure the quality of the imaging in the fluorescence signal acquisition system and facilitate subsequent experimental processes, such as identification, counting, and analysis, it is necessary to test and calibrate the imaging system using a fluorescent dye. Prepare solutions of appropriate concentrations of FAM and HEX fluorophores and inject the solutions into the chambers of the chip, ensuring that all chambers are filled. Capture images of the chip using the fluorescence signal acquisition system in a darkroom, with an exposure time set to 1 s. Utilize ImageJ software to read and analyze the captured fluorescence images. Extract the central point positions from the fluorescence images, draw horizontal and vertical lines through the central points as benchmarks for testing and calibration, and analyze image uniformity.

#### 2.7.3. Linearity of the Fluorescence Signal Acquisition System

To verify the linearity of the fluorescence signal acquisition system, which refers to the system’s dynamic range being within the required range, the signal range of the emitted fluorescence intensity on the microarray chip after photodetection by the detector was measured. The coefficient factor R^2^ after linear fitting was used. The standard microarray chip, as shown in Figure 3, is uniformly distributed with many fluorescent spots arranged in a matrix pattern by rows and columns. Each column of fluorescent spots has the same concentration, while the concentration of fluorescent spots increases linearly from left to right within each row. Signal intensity at any randomly selected point within each column of fluorescent spots is measured as a single measurement value for that concentration. The linearity of the imaging system of the detection device is evaluated through linear fitting of R^2^ values obtained from these measurements. The density of fluorophores in each column is shown in Table 2.

#### 2.7.4. Identification of *H. pylori* Drug Resistance by the E-Test Method

For the obtained *H. pylori* strains, clarithromycin-resistance testing was conducted using MIC test strips (MTS) purchased from Liofilchem (Roseto degli Abruzzi, Italy). After 48 h of growth on plates, isolates were suspended in sterile saline and adjusted to a turbidity of 0.5 McFarland (1 × 10^8^~2 × 10^8^ CFU/mL) according to Clinical and Laboratory Standards Institute (CLSI) recommendations. Bacterial suspension was swabbed onto MH agar medium using a sterile cotton swab. Plates with E-test strips for clarithromycin were then incubated for 48 h under microaerophilic conditions. According to the guidelines of the European Committee on Antimicrobial Susceptibility Testing (EUCAST), bacterial isolates were considered resistant to the antibiotic if the clarithromycin MIC value was >0.5 μg/mL [23].

#### 2.7.5. Microarray Chip Detection

The obtained Helicobacter pylori strains underwent genomic DNA extraction using the Rapid Bacterial Genomic DNA Isolation Kit (Bioteke Corporation, B518225, Shanghai, China). The extracted Helicobacter pylori genomic DNA samples were mixed with upstream primers, downstream primers, and probes, along with other necessary components for PCR. The PCR reaction mixture was prepared as follows: final concentrations of 10 μM for each upstream and downstream primer, 1 μL of Helicobacter pylori genomic DNA sample, 10 μL of 2× ChamQ Universal SYBR qPCR Master Mix (Nanjing Novogene Biotechnology Co., Ltd., Q441-02, Nanjing, China), and ultrapure water added to make the reaction volume 20 μL. The prepared PCR reaction mixture was injected into the pre-printed chip, with the gel point array arranged as shown in Figure 4. PCR cycling conditions were set as follows: initial denaturation at 95 °C for 30 s, followed by 40 cycles of denaturation at 95 °C for 3 s, annealing and extension at 60 °C for 10 s. After PCR, the chip underwent a hybridization process at 50 °C for 30 min. Post-hybridization, the chip was washed with 1× SSPE, and fluorescence signals were observed using the designed fluorescence signal acquisition system.

## 3. Results and Discussion

### 3.1. Verification of Excitation Light Path and Imaging System Performance

#### 3.1.1. Verification of Excitation Light Path Performance

In ZEMAX non-sequential mode, a radial light source with an energy of 1 W was set. A rectangular detector of 12 mm × 12 mm was placed 65 mm away from the light source, as shown in Figure 5a. The total power detected by the detector was 0.992 W, achieving a light source utilization efficiency of 99.2%. The light spot distribution was uniform, with an average illuminance of 0.882 W/cm^2^, as depicted in Figure 5b,c.

#### 3.1.2. Verification of Imaging System Performance

The modulation transfer function (MTF) curve is a primary method for comprehensively evaluating the imaging quality of optical systems, as shown in Figure 6a. At the camera cutoff frequency of 143 lp/mm, the MTF values for the entire field of view are all above 0.4, and the MTF curves exhibit consistent trends. This indicates good consistency in imaging quality across different fields of view and meets the performance requirement of MTF ≥ 0.3 as per the system design. Additionally, the MTF curves in the meridional direction overlap with those in the sagittal direction, indicating that the optical system has no astigmatism in imaging. Furthermore, from Figure 6b, it can be observed that the RMS achieves its maximum value (RMSmax = 1.794 μm) at IMA of −8.798 mm. This indicates that the optical imaging system has excellent signal acquisition capabilities across all fields of view, providing uniformly good illumination for images. The imaging system exhibits characteristics such as a simple structure, low aberrations, and good imaging quality.

### 3.2. Optical and Thermal Characteristics of Chip Materials

#### 3.2.1. Transparency Characteristics of Chip Material

Using COMSOL, the transmittance of the COC material was simulated and solved based on the wavelength of the excitation light source for the fluorescent dye (497 nm). A radiation light source of 1 W was used to instantaneously illuminate the material, and the transmittance of the model was calculated through the transmitted radiation power. The simulation results are shown in Figure 7.

Simulation and modeling were conducted for the eight materials listed in Table 3. The transmittance for all eight materials was found to be satisfactory, meeting the design requirements for the chip, as shown in Table 4.

#### 3.2.2. Thermal Characteristics of Chip Material

The PCR process is a rapid-temperature-cycling process ranging from 4 to 100 °C. Finite element simulation of the thermal deformation of chip material using COMSOL software mainly involves six parameters: thermal conductivity, density, coefficient of thermal expansion, constant-pressure heat capacity, Young’s modulus, and Poisson’s ratio, as listed in Table 3.

To conduct a coupled thermal and stress analysis, the heat transfer boundary conditions and mechanical boundary conditions were directly applied to the finite element model. The node temperatures and displacements were treated as unknown variables for solving. The X-axis was set along the longer edge of the chip, the Y-axis along the shorter edge, and the Z-axis along the normal direction of the chip plane. The displacements in the XYZ directions were measured to evaluate whether the selected materials meet the requirements. Figure 8 presents the 2D plots of XYZ displacements for the eight materials. The vertical axis represents the coordinate distance from the reference point, and the horizontal axis represents the arc length change after slight deformation. The three lines, u, v, and w, represent the displacement change curves in the x, y, and z directions, respectively. The blue line corresponds to the x-direction displacement, the green line to the y-direction displacement, and the red line to the z-direction displacement. Based on the variations in each direction shown in the figure, the maximum displacement change occurred in the z-direction. The z-direction displacement changes for the eight materials were as follows: ABS: −1.35; COC: −0.64; PC: −0.83; PMMA: −0.60; PP: −0.53; PS: −0.84; PTFE: −1.13; and PVDF: −0.50.

For the calculated results of the deformation variables for the eight materials, please refer to Table 4. The maximum deformation variable that the chip will undergo during the PCR reaction process is calculated. Since the thickness of the chip is only 0.7 mm, the deformation variable has a significant impact on the later signal collection process. Therefore, this indicator is crucial for selecting chip materials. A too large deformation variable can lead to uneven filling of the chamber with liquid, causing inconsistency in the reactions of individual units in the array. From the results, materials such as ABS, PC, PS, and PTFE are excluded as their absolute values of deformation variables in the z-direction exceed 0.7 mm.

#### 3.2.3. Material Photoexcitation Effects

Based on the fluorescence collection results of the fluorescence collection system, the fluorescence effects of four materials were analyzed using ImageJ, as illustrated in Figure 9. In Figure 9A, the fluorescence mean value for COC material is 2000; in Figure 9B, the fluorescence mean value for PMMA material is 19,800; in Figure 9C, the fluorescence mean value for PP material is 3100; and in Figure 9D, the fluorescence mean value for PVDF material is 4600. COC material exhibits the lowest degree of excitation and the smallest background value. Consequently, COC was chosen as the substrate material for the chip in subsequent experiments.

### 3.3. Device Performance Verification

#### 3.3.1. Light Uniformity

To prepare solutions of FAM fluorophore and HEX fluorophore at a specific concentration, inject the solution into the cavity chip, ensuring that all chambers are filled. Capture images of the chip in a darkroom using the fluorescence signal acquisition system, with an exposure time set to 1 s. Utilize ImageJ software to read and analyze the acquired fluorescence images. Extract the center point position from the collected fluorescence images, draw horizontal and vertical lines passing through the center point as reference lines for testing and calibration and analyze the uniformity of the images. The grayscale curves of the horizontal and vertical lines for FAM fluorophore are shown in Figure 10A,B. The mean grayscale values are 34,300.562 and 33,564.990, respectively. In Figure 10A, the non-edge portion exhibits stable oscillations distributed between grayscale values of 31,000 and 36,000, with a relative range of 14%. In Figure 10B, the non-edge portion shows stable oscillations distributed between grayscale values of 31,000 and 35,500, with a relative range of 13%. The grayscale curves of the horizontal and vertical lines for HEX fluorophore are depicted in Figure 10C,D. The mean grayscale values are 31,185 and 30,524.299, respectively. In Figure 10C, the non-edge portion shows stable oscillations distributed between grayscale values of 28,000 and 31,500, with a relative range of 11%. In Figure 10D, the non-edge portion exhibits stable oscillations distributed between grayscale values of 30,000 and 31,500, with a relative range of 1%. Therefore, it is concluded that the fluorescence signal acquisition system demonstrates good uniformity calibration, meeting the requirements for subsequent experiments.

#### 3.3.2. Linearity

After the system is powered on and stabilized, a standard microarray chip is scanned. The fluorescence intensities of the fluorophore clusters are obtained using the background subtraction method, and the data are presented in Table 5. The logarithm of the fluorophore cluster density is taken as the x-axis, and the logarithm of the fluorescence intensity of the fluorophore clusters is taken as the y-axis for plotting. Linear fitting is performed on the data in Table 5, yielding a linear correlation coefficient of R^2^ = 0.9513, as illustrated in Figure 11.

### 3.4. Detection of Helicobacter pylori Drug-Resistance Genes

The samples consisted of 19 strains of *H. pylori* with clarithromycin-resistant mutation genes. The E-test detection kit was used to determine the minimum inhibitory concentration (MIC), as shown in Table 6. Simultaneously, the corresponding nucleic acid samples were tested using the microarray chip detection method. In the E-test, all 19 strains showed MIC results greater than 0.5 μg/mL, and these results were 100% consistent with the microarray chip detection results. The detected mutation sites of the drug-resistant genes are presented in Table 6.

Standard procedure for the E-test includes grinding and homogenizing gastric tissue samples, isolating and purifying strains, inoculating strains, placing E-test strips, and incubating cultures (16~24 h, typically 48 h for *H. pylori*) [35]. Finally, the minimum inhibitory concentration (MIC) of the tested drug against the bacterial strain is determined by visually observing the intersection of the inhibition zone with the corresponding scale on the E-test strip.

In comparison with conventional E-test detection, the method established in this study enables direct detection of genes specific to *H. pylori*, bypassing the need for complex bacterial culture processes, thereby shortening the detection time to within 3 h (including nucleic acid extraction). Although a conventional E-test can also identify resistant drug types, each E-test can only detect one type of resistance. To detect other resistances, it requires replacing E-test strips. However, this method allows for adjustment of primers and probes targeting different resistance gene mutations, achieving simultaneous detection of multiple resistance genes. Conventional E-tests cannot determine the mutation sites of resistant genes after detecting resistance, thus they are unable to precisely guide drug usage. In contrast, this method accurately detects resistance caused by different gene mutations, providing more targeted guidance for drug use. In summary, this method offers advantages over a traditional E-test in terms of speed, convenience, and accuracy.

## 4. Conclusions

A microarray chip designed for the detection of drug-resistant genes in *H. pylori* was developed using SolidWorks software. Finite element analysis of the crucial material’s optical and thermal properties influencing chip quality was conducted using COMSOL software. Among the eight materials analyzed (ABS, COC, PC, PMMA, PP, PS, PTFE, and PVDF), the optical properties of ABS, COC, PC, PMMA, PP, PS, PTFE, and PVDF all met experimental requirements. However, ABS, PC, PS, and PTFE were excluded due to significant deformation during heat extraction. Among the remaining four materials (COC, PMMA, PP, and PVDF), COC exhibited the smallest background fluorescence when excited by FAM light, making it the chosen material for chip processing. Comparative detection experiments on *H. pylori* drug resistance were conducted. The results demonstrated that the microarray chip detection method established in this study, which analyzed and detected 19 clarithromycin-resistant mutation sites, significantly reduced the detection period and saved manpower and resources compared to traditional clinical detection methods. Current molecular biology detection methods, both domestically and internationally, only target the A2143G single mutation site or the A2142G, A2143G, A2142C, three clarithromycin-resistant mutation sites. However, in practical applications, there is a risk of overlooking *H. pylori* samples with clarithromycin resistance. The current assay in this study is only for the detection of mutated genes for *H. pylori* resistance to clarithromycin, and on the basis of the principles underlying this study, it is possible to construct assays for the production of mutations by other drugs (including nitroimidazoles, macrolides, penicillins, tetracyclines, and rifamycins, among others). Therefore, the microarray chip sensor established in this study comprehensively studied and explored 19 clarithromycin-resistant mutation genes in *H. pylori*, providing a more extensive coverage of detection sites.

## Figures and Tables

**Figure 1 micromachines-15-00416-f001:**
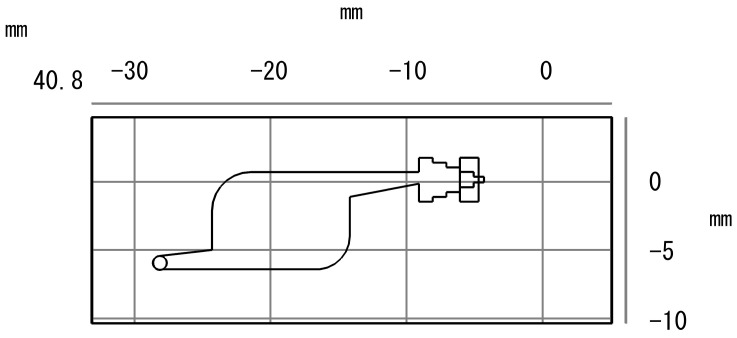
Microarray chip design.

**Figure 2 micromachines-15-00416-f002:**
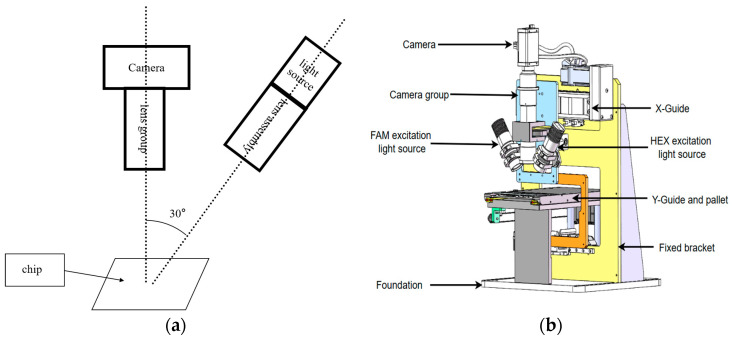
Schematic diagram of the detection system: (**a**) schematic diagram of device positions; (**b**) schematic diagram of system structure.

**Figure 3 micromachines-15-00416-f003:**
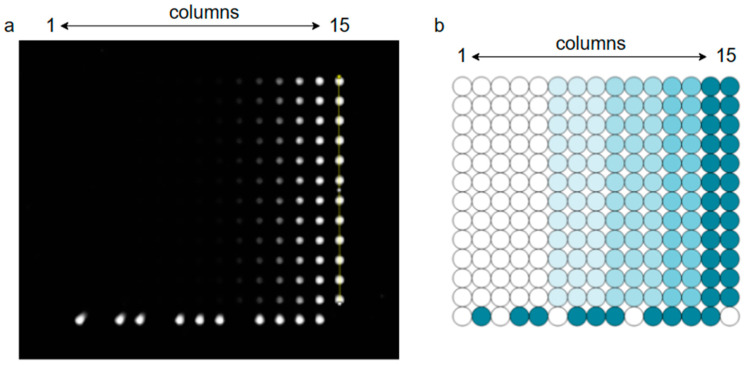
Standard microarray chip image: (**a**) physical drawing of a standard microarray chip (The brighter the circle, the higher the concentration of fluorophores); (**b**) schematic diagram of a standard microarray chip (The darker the circle, the higher the concentration of fluorophores).

**Figure 4 micromachines-15-00416-f004:**
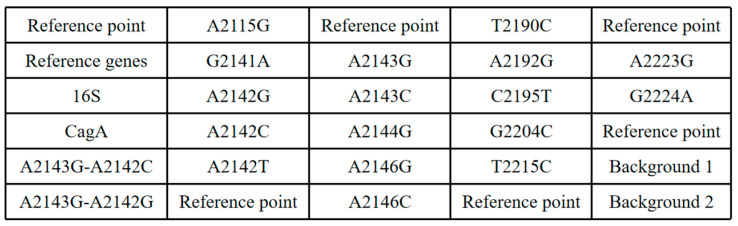
Gel array layout. (Note: Positioning points are used for the auxiliary positioning of this chip during the imaging recognition process to ensure the correct position of the chip; primers and probes for the internal reference genes are shown in the IC section (IC-F, IC-R, IC-P) of Table 1 and are used to assess the validity of this test (or whether the nucleic acid samples are detected in this experiment); background points are used to calculate the signal-to-noise ratio during the analysis process and to determine the negatives and positives.).

**Figure 5 micromachines-15-00416-f005:**
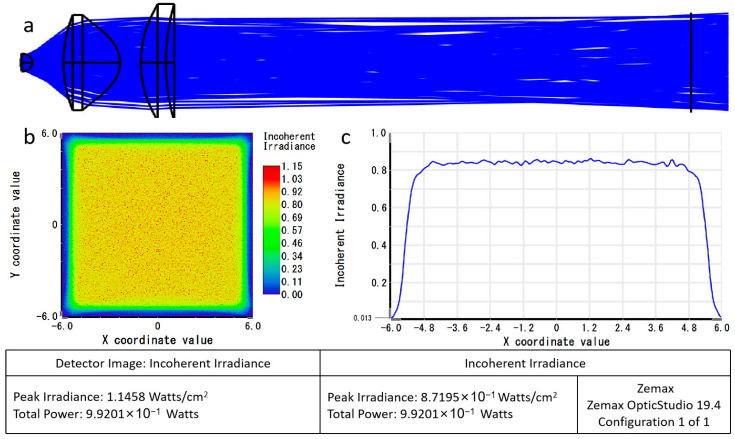
Simulation of excitation light path performance: (**a**) collimated light diagram; (**b**) received power diagram; (**c**) average illuminance diagram.

**Figure 6 micromachines-15-00416-f006:**
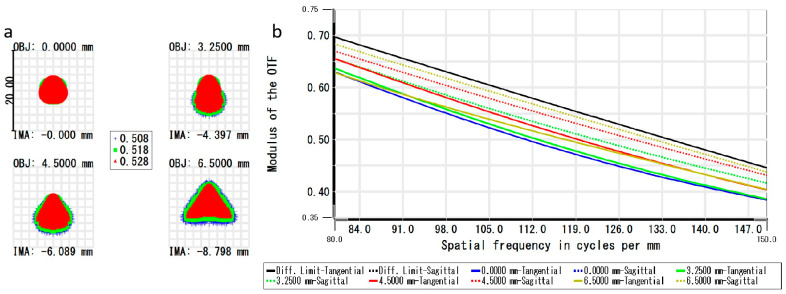
Simulation of imaging system performance: (**a**) MTF curve diagram (Different colours indicate different wavelengths of light); (**b**) airy disk diagram (Diff. Limit-Sagittal coincides with Diff. Limit-Tangential; 0.0000 mm-Sagittal coincides with 0.0000 mm-Tangential).

**Figure 7 micromachines-15-00416-f007:**
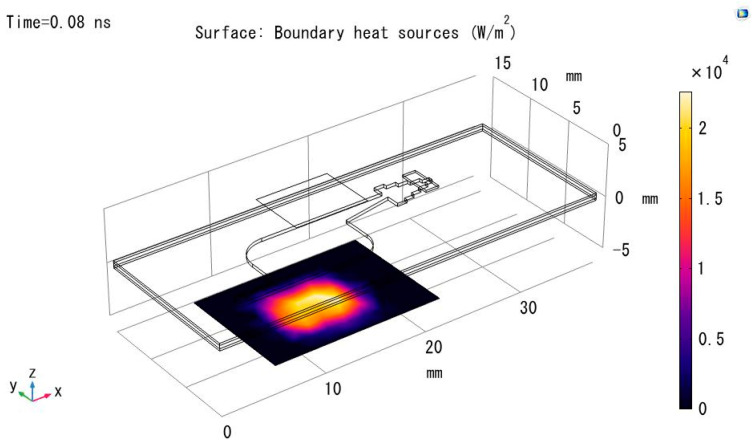
Simulation of light transmittance power for COC material.

**Figure 8 micromachines-15-00416-f008:**
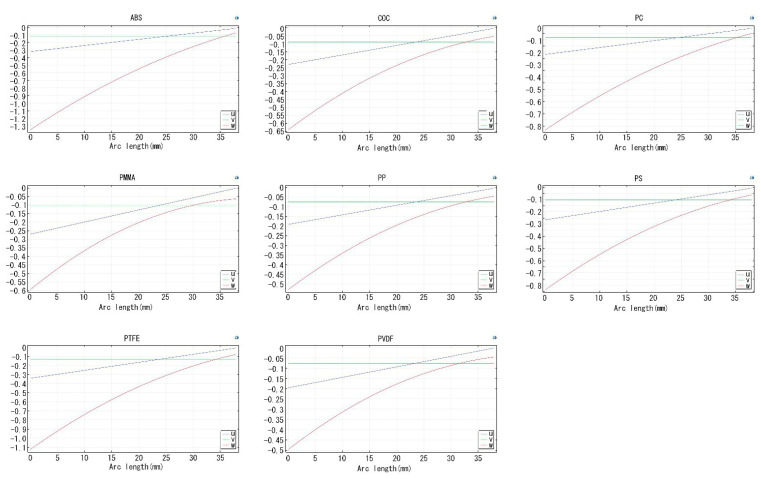
Material displacement and volume strain maps.

**Figure 9 micromachines-15-00416-f009:**
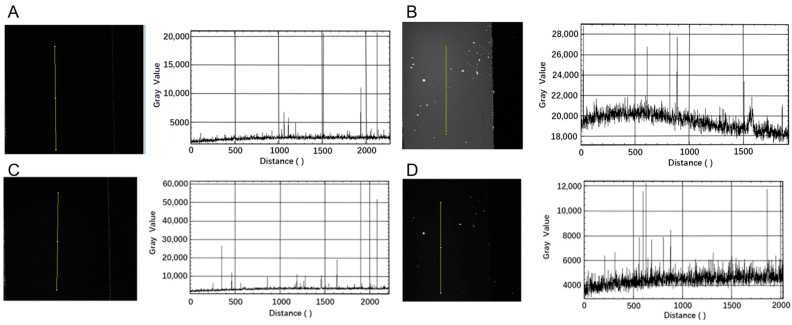
Mean fluorescence intensity of four materials under excitation (The yellow line shows the ImageJ detection position information, and the upper part is the Distance 0): (**A**) excitation image of COC; (**B**) excitation image of PMMA; (**C**) excitation image of PP; (**D**) excitation image of PVDF.

**Figure 10 micromachines-15-00416-f010:**
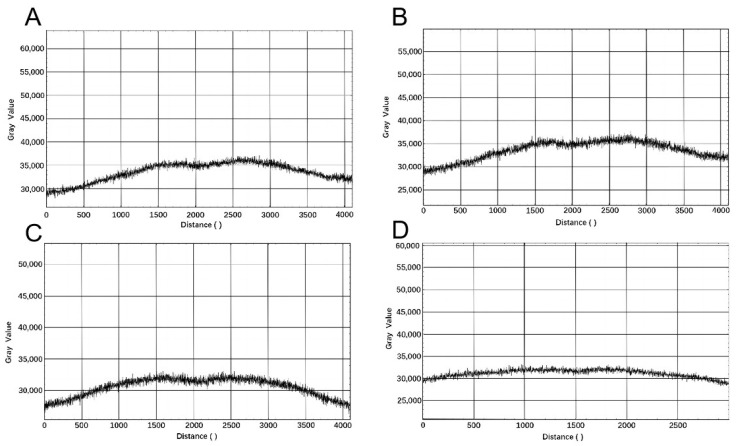
Light uniformity chart of the fluorescence signal acquisition system: (**A**) transversal intensity of FAM light source; (**B**) longitudinal intensity of FAM light source; (**C**) transversal intensity of HEX light source; (**D**) longitudinal intensity of FAM light source.

**Figure 11 micromachines-15-00416-f011:**
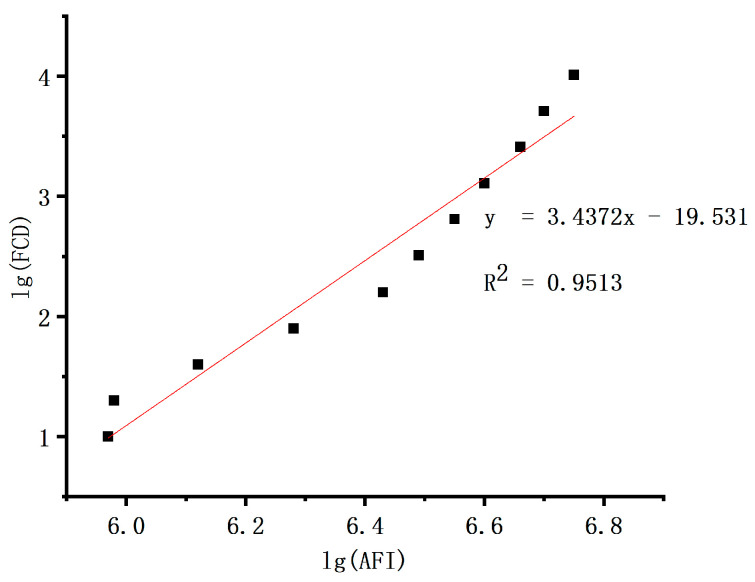
Linear regression curve of the fluorescence signal acquisition system.

**Table 1 micromachines-15-00416-t001:** Primer and probe table.

Primers	Sequence (5′-3′)	Corresponding Site(s)
16s rRNA-F	CTCATGCGAAGGCGACCT	16s rRNA
16s rRNA-R	TCTAATCCTGTTTGCTCCCCA	16s rRNA
16s rRNA-P	FAM-ATTACTGACGCTGATTGCGCGAAAGC-MGB	16s rRNA
CagA-F	ATAATGCTAAATTAGACAACTTGAGCGA	CagA
CagA-R	TTAGAATAATCAACAAACATCACGCCAT	CagA
CagA-P	FAM-TCAGCTAGCCCTGAACCCATTTACGCTAC-MGB	CagA
A2115G-F	CAGTGAAATTGTAGTGGAGGTGATG	A2115G
G2141A-F	ACCCGCGGCAAGACAA	G2141A
A2142G-F	CCGCGGCAAGACTGG	A2142G
A2142C-F	CCGCGGCAAGACAGC	A2142C
A2142T-F	ACCCGCGGCAAGACAGT	A2142T
A2143G-F	CCGCGGCAAGACGTAG	A2143G
A2143C-F	CCCGCGGCAAGACGTAC	A2143C
A2144G-F	GCGGCAAGACGGATGG	A2144G
A2146G-F	GCGGCAAGACGGAAAAG	A2146G
A2146C-F	CGGCAAGACGGAACGC	A2146C
T2190C-F	CAACTTAGCACTGCTAATGGGAATAC	T2190C
A2192G-F	ACTTAGCACTGCTAATGGGAATATCATCT	A2192G
C2195T-F	CTTAGCACTGCTAATGGGAATATCATCT	C2195T
G2204C-F	GGAATATCATGCGCAGGATGC	G2204C
T2215C-F	AGGATAGGTGGGAGGCCC	T2215C
A2223G-F	GTGGGAGGCTTTGAAGTCG	A2223G
G2224A-F	GGTGGGAGGCTTTGAAGTACA	G2224A
A2143G-A2142G-F	CCGCGGCAAGACGGGG	A2143G-A2142G
A2143G-A2142C-F	CCGCGGCAAGACGGGG	A2143G-A2142C
R-1	GGCTTTGGCTCTTATGGAGC	A2115G; G2141A; A2142G; A2142C; A2142T; A2143G; A2143C; A2144G; A2146G; A2146C;
R-2	GGGTGGTATCTCAAGGATGGCT	T2190C; A2192G; C2195T; G2204C; T2215C; A2223G; G2224A
P1	FAM-CCGTGGACCTTTACTACAA-MGB	A2115G; G2141A; A2142G; A2142C; A2142T; A2143G; A2143C; A2144G; A2146G; A2146C
P2	FAM-ATAGGTGGGAGGCTTT-MGB	T2190C; A2192G; C2195T
P3	FAM-CTTTGGCTCTTATGGAG-MGB	G2204C; T2215C; A2223G; G2224A
IC-F	CTGGAGCTAGGCATGATTGGA	Reference gene (IC)
IC-R	CACATTGTTGCCTTGTTGGTCTTT	Reference gene (IC)
IC-P	FAM-ACGGTGGCGTTCCAATCA-MGB	Reference gene (IC)

**Table 2 micromachines-15-00416-t002:** Density distribution of fluorophores in each column of a standard microarray chip.

Columns	Fluor per μm^2^	μM Labeled
15	40,960	8.14778
14	20,480	4.07389
13	10,240	2.03695
12	5120	1.01847
11	2560	0.50924
10	1280	0.25462
9	640	0.12731
8	320	0.06365
7	160	0.03183
6	80	0.01591
5	40	0.00796
4	20	0.00398
3	10	0.00199
2	5	0.00099
1	0	0

**Table 3 micromachines-15-00416-t003:** Material modulus table.

Material Category	Thermal Conductivity (W/m/K)	Density (g/cm^3^)	Coefficient of Thermal Expansion	Constant-Pressure Heat Capacity (J/(kg × K))	Young’s Modulus (Mpa)	Poisson’s Ratio
COC	0.1200	1.0200	0.7000 × 10^−4^	1290	3200	0.3900
PVDF	0.1000	1.7800	0.6000 × 10^−4^	1170	1030	0.4000
PMMA	0.1920	1.1500	0.8300 × 10^−4^	1465	3160	0.3200
PP	0.1470	0.8900	0.5800 × 10^−4^	1881	890	0.4203
PC	0.1975	1.2000	0.6530 × 10^−4^	1172	2320	0.3902
PS	0.0800	1.0500	0.8000 × 10^−4^	1300	3000	0.3870
PTFE	0.2560	2.1000	1.0300 × 10^−4^	1000	1140	0.4000
ABS	0.2256	1.0200	0.9500 × 10^−4^	1386	2000	0.3940

**Table 4 micromachines-15-00416-t004:** Statistical summary of simulation computational results.

Materials	Deformation (mm)			Optical Transmittance
	X	Y	Z	
ABS	−0.32	−0.12	−1.35	0.82
COC	−0.23	−0.09	−0.64	0.84
PC	−0.22	−0.08	−0.83	0.84
PMMA	−0.27	−0.11	−0.6	0.86
PP	−0.19	−0.07	−0.53	0.86
PS	−0.26	−0.1	−0.84	0.85
PTFE	−0.34	−0.13	−1.13	0.91
PVDF	−0.2	−0.08	−0.5	0.89

**Table 5 micromachines-15-00416-t005:** Linearity data of the fluorescence signal acquisition system.

	Spot Area	AFI (Actual Fluorescence Intensity)	lg (AFI)	FCD (Fluorophore Cluster Density)	lg (FCD)
1	661	5.62 × 10^6^	6.75	10,240	4.01
2	661	5.07 × 10^6^	6.70	5120	3.71
3	661	4.56 × 10^6^	6.66	2560	3.41
4	661	3.95 × 10^6^	6.60	1280	3.11
5	661	3.52 × 10^6^	6.55	640	2.81
6	661	3.08 × 10^6^	6.49	320	2.51
7	661	2.66 × 10^6^	6.43	160	2.20
8	661	1.90 × 10^6^	6.28	80	1.90
9	661	1.32 × 10^6^	6.12	40	1.60
10	661	9.56 × 10^5^	5.98	20	1.30
11	661	9.32 × 10^5^	5.97	10	1.00
Background Points	661	7.40 × 10^5^	/	/	/

**Table 6 micromachines-15-00416-t006:** Comparative detection experiment table.

Sample ID	E-Test	Fluorescence Intensity								Area	Genetic Mutation Locus
		16S rRNA Mean	CagA Mean	Reference Gene Mean	Mutant Gene Mean	Mutant/ BackM	Back Ground Value 1	Back Ground Value 2	Back Ground MEAN		
8	32	247.94	251.02	249.37	252.48	4.25	77.84	41.03	59.44	124	A2143C
1	16	249.70	246.82	249.29	245.95	3.85	63.87	63.91	63.89	124	A2413G
15	8	254.06	254.63	254.98	204.82	3.07	49.93	83.56	66.74	124	A2115G
14	24	230.07	224.69	232.40	219.81	4.02	49.82	59.65	54.73	124	A2143G
11	24	247.15	243.77	242.63	248.09	3.93	74.98	51.33	63.15	124	A2143G
3	96	247.89	248.88	249.11	253.10	6.40	39.90	39.24	39.57	124	A2115G
					241.09	6.09					C2195T
5	12	253.81	254.34	254.17	251.57	3.41	69.72	77.64	73.68	124	A2115G
2	8	247.89	251.99	252.78	234.74	3.32	68.77	72.77	70.77	124	A2115G
9	16	253.10	252.78	248.70	245.79	3.80	63.25	66.04	64.65	124	A2143G
13	16	212.57	224.69	233.99	177.06	3.66	50.71	45.98	48.35	124	A2143G
10	32	248.70	245.84	249.70	232.19	4.38	42.78	63.22	53.00	124	A2143G
17	16	250.37	251.19	250.28	253.85	3.66	70.98	67.69	69.34	124	A2143G
18	64	253.84	254.27	252.70	251.72	5.48	40.32	51.57	45.94	124	A2142G
6	16	251.57	252.46	252.37	250.01	3.66	67.34	69.43	68.38	124	A2143G
16	24	254.04	252.19	254.88	249.87	3.98	48.79	76.81	62.80	124	A2142G
12	32	251.52	248.54	247.22	245.11	4.10	56.02	63.42	59.72	124	A2143G
19	8	253.40	253.46	253.84	188.90	2.60	73.50	71.68	72.59	124	A2115G
4	6	253.95	254.32	253.52	183.54	2.50	73.19	73.68	73.44	124	A2115G
7	1.5	252.90	249.10	251.02	120.53	1.61	70.93	79.05	74.99	124	A2115G

## Data Availability

The data that support the findings of this study are available from the corresponding author upon reasonable request.

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
