# Peer review of "Research on Key Technologies of Microarray Chips for Detecting Drug-Resistant Genes in Helicobacter pylori"

_micromachines, 2024, doi:10.3390/mi15030416_

Round 1

Reviewer 1 Report

Comments and Suggestions for Authors

The Authors propose a microarray chips for detecting drug-resistant genes in helicobacter pylor. Experimental results have been reported. The manuscript is well written and it deserves the publication, after addressing the following comments:

1. -          In the Introduction, to help the reader in the performance understanding, the Authors could also stress the benefits of detecting bacteria, as Helicobacter or E.coli, as the facing of Anti-Microbial Resistance. A great research effort has been spent during the last years to propose low cost, very sensitive and fast biosensors to drive the choice of the antibiotic, also for biofilm suppression (see, i.e., “Novel micro-nano optoelectronic biosensor for label-free real-time biofilm monitoring.” Biosensors, 11(10), 361, 2021;  “Gram-type differentiation of bacteria with 2D hollow photonic crystal cavities.” Applied Physics Letters, 113(11), 111101, 2018; “Attachment and antibiotic response of early-stage biofilms studied using resonant hyperspectral imaging.” NPJ biofilms and microbiomes, 6(1), 2020; “A Novel Hybrid Platform for Live/Dead Bacteria Accurate Sorting by On-Chip DEP Device,” International Journal of Molecular Sciences, 24(8), 7077, 2023; “A review on impedimetric immunosensors for pathogen and biomarker detection.” Medical microbiology and immunology, 209(3), 343-362, 2020).

- More details about the chip design and constraints have to be reported. In particular, focus how the cavity, substrate and top layer thickness affect the performance.

- The quality of all Figures have to be improved, as Fig. 2, Fig. 6, Fig. 7, Fig. 8, Fig. 9, Fig. 10. All figures should be in English and easily readable.

- The last section should be entitled "Conclusions and discussions". The main results should be compared with the state-of-the-art.

Author Response

Thank you for your letter and for the reviewers’ comments concerning our manuscript entitled “Research on Key Technologies of Microarray Chips for Detecting Drug-Resistant Genes in Helicobacter pylor” (micromachines-2809177). Those comments are all valuable and very helpful for revising and improving our paper, as well as the important guiding significance to our researches. We have studied comments carefully and have made correction which we hope meet with approval. Revised portion are marked in red in the paper. The main corrections in the paper and the responds to the reviewer’s comments are as flowing.

Response 1: Special thanks to you for your good comments. Based on the reviewers' suggestions, we have added this section. Your valuable comments are especially appreciated. These changes will not affect the content and framework of the paper. The changes have been marked in red in the revised draft. They are located on page 3, line 115 of the revised draft.

Response 2: We have rechecked this section in response to the reviewer's suggestions and have replaced the images with clearer ones. Special thanks to you for your good comments. These changes will not influence the content and framework of the paper.

Response 3: Special thanks to you for your good comments. Based on the reviewers' suggestions, we have added this section. Your valuable comments are especially appreciated. These changes will not affect the content and framework of the paper. The changes have been marked in red in the revised draft. They are located on page 14, line 364 and line 393, of the revised draft.

Reviewer 2 Report

Comments and Suggestions for Authors

I have reviewed the manuscript by Guo et al., “Research on key technologies of microarray chips for detecting drug-resistant genes in Helicobacter pylori.“ This paper describes the design of a microarray chip for detecting drug-resistance genes in a pathogenic bacterial species. They have identified COC as the promising material for chip fabrication and conclude that this chip-based method aligns well with the established E-test detection kit. 

This manuscript is submitted when the hybridization-based microarray method is gradually fading due to its technical sensitivity limitations, cost, and operational complexity and when the sequencing-based method is rising. However, a reliable and easy-to-adopt method for detecting drug-resistant H. pylori still has scientific and applied value.

I am surprised to see the spelling error of the bacterial species name in the manuscript title. Following international convention, it is essential to present biological species names in italics.  

The introduction of this article needs to be better articulated. It is worth mentioning that the Microarray chips method did not exist in the mid-1980s due to a lack of sequence data at that time. 

In the materials and method section, figures 3 and 4 are poorly explained in the legend. The message in fig 3 needs to be explained. Also, in fig 4, there needs to be an explanation about the reference point and the reference genes. Most importantly, how the PCR reaction is integrated into the microarray detection technology must be clarified. The readers (researchers) must understand this to adopt it in their laboratories. The genomic DNA isolation kit and different molecular biology reagents do not have information about their purchase source (for example, company name, catalog number, and country).

I was surprised not to find a discussion section in this article. The authors are encouraged to combine results and discussion in one section, with proper caveats.

Section 3.4, “Detection of Helicobacter pylori drug resistance genes,” is a critical component of this manuscript that requires elaboration to support the conclusion. It is unclear how this method will advance drug-resistance gene detection in this pathogenic bacteria.

Comments on the Quality of English Language

Minor

Author Response

Thank you for your letter and for the reviewers’ comments concerning our manuscript entitled “Research on Key Technologies of Microarray Chips for Detecting Drug-Resistant Genes in Helicobacter pylor” (micromachines-2809177). Those comments are all valuable and very helpful for revising and improving our paper, as well as the important guiding significance to our researches. We have studied comments carefully and have made correction which we hope meet with approval. Revised portion are marked in red in the paper. The main corrections in the paper and the responds to the reviewer’s comments are as flowing.

Response 1: We have rechecked this section in response to the reviewer's suggestions. Special thanks to you for your good comments. These changes will not influence the content and framework of the paper.

Response2: Special thanks to you for your good comments. Based on the reviewers' suggestions, we have added this section. Your valuable comments are especially appreciated. These changes will not affect the content and framework of the paper. The changes have been marked in red in the revised draft. They are located on page 2, line 60, of the revised draft.

Response 3: Special thanks to you for your good comments. Based on the reviewers' suggestions, we have added this section. Your valuable comments are especially appreciated. These changes will not affect the content and framework of the paper. The changes have been marked in red in the revised draft. They are located on page 6, line 187, and page 7, line 217, of the revised draft.

Response 4: Special thanks to you for your good comments. Based on the reviewers' suggestions, we have added this section. Your valuable comments are especially appreciated. These changes will not affect the content and framework of the paper. The changes have been marked in red in the revised draft. They are located on page 14, line 364 and line 393, of the revised draft.

Response 5: Special thanks to you for your good comments. Based on the reviewers' suggestions, we have added this section. Your valuable comments are especially appreciated. These changes will not affect the content and framework of the paper. The changes have been marked in red in the revised draft. They are located on page 14, line 364 of the revised draft.

Round 2

Reviewer 1 Report

Comments and Suggestions for Authors

The Authors have modified the manuscript according to the Reviewer suggestions. However to deserve the publication a wide overview of the competing technologies have to be reported as pointed out in the comment 1 of the previous revision

Reviewer 2 Report

Comments and Suggestions for Authors

This manuscript requires several editorial corrections and should include specific missing information, as detailed below.

  1. In the manuscript title, "i" is missing in the spelling of Helicobacter pylori.
  2. The species name can be spelled as H. pylori after mentioning the full name once in the introduction.
  3. Lines 51-61: It is crucial to elaborate on the role of microarray chips as key technological components in proteomics. How exactly do they contribute to the analysis of protein expression patterns? 
  4. Fig 3 and Table 2: I do not see any column number annotated in Fig 3 that will correspond to Table 2. 
  5. Lines 215-216 states, "Internal reference genes are shown in the IC section of Table 1". Where is the IC section in Table 1? I do not see a connection between Table 1 and Figure 4.
  6. Lines 220-223: As I mentioned earlier, please clearly describe how the PCR method is connected with the microarray chip in the protocol. Are you PCR amplifying the isolated DNA before hybridization to the chip? Please clarify.
  7. Lines 360-369: H. pylori in italics. How the chip-based method is better than the E-test needs to be clarified. Is it faster, cheaper, more sensitive, or more accurate?
Comments on the Quality of English Language

Round 3

Reviewer 1 Report

Comments and Suggestions for Authors

The Authors have modified the manuscript accordingly. However, the literature should be improved by citing the most relevant papers in the framework.

Round 4

Reviewer 1 Report

Comments and Suggestions for Authors

The Authors have modified the manuscript according to the Reviewer suggestions.